# Direct detection of SABRE-SHEATH hyperpolarization and spin-lattice relaxation of [1-$^{13}$C]pyruvate
John Z. Myers [1] ✉, Markus Plaumann[2], Kai Buckenmaier[3], Andrey N. Pravdivtsev [4] & Rainer Körber [1] ✉

Nuclear magnetism is typically investigated by perturbing the spin system with radio frequency pulses, but low polarization and detection using induction coils limit direct access to the longitudinal magnetization. The hyperpolarization technique SABRE-SHEATH requires ultra-low magnetic fields for spin order transfer; consequently, SQUID sensors with a frequency-independent sensitivity are well-suited for unperturbed detection in this regime. We demonstrate direct observation of hyperpolarization build up ($T_B$) and spin lattice relaxation ($T_1$) in [1-$^{13}$C]pyruvate, hyperpolarized with SABRE-SHEATH at 150 nT and 500 nT. The values for $T_B$ of 36 s and 26 s and $T_1$ of 40 s and 43 s, respectively, suggests a shift in dominant polarization transfer efficacy or complexes, highlighting the method's merit in characterizing hyperpolarization pathways. Moreover, as demand for hyperpolarized probes in metabolic imaging continues to grow, the exceptional time resolution makes direct detection a valuable tool for understanding and optimizing polarization dynamics and reactor designs.

Nuclear magnetization is an important phenomenon, indispensable to many fields, including applications to medical resonance imaging (MRI)[1], including non-invasive metabolic probing for cancer diagnosis and treatment assessment[2], and use in the detection and quantification of contaminants in food[3]. To support the further development of these applications, it is of interest to develop techniques to better characterize the behaviour of nuclear magnetism under various conditions.

Underpinning many nuclear magnetism studies, nuclei of interest commonly have a spin of 1/2 and thus, two energy states, separated by their Zeeman energy: $\Delta E = \hbar\gamma B_0$, where $\hbar$ is the reduced Planck constant, $\gamma$, the gyromagnetic ratio of the nuclei and $B_0$, a magnetic field the nuclei are exposed to[4]. At thermal equilibrium, which is the most common state probed, the relative populations of these two energy states are given by the Boltzmann distribution $N_\alpha/N_\beta = \exp\left[-\Delta E/(k_B T)\right]$, where $N_\alpha$ and $N_\beta$ refer to the number of spins in states $\alpha$ and $\beta$, respectively. $k_B$ refers to the Boltzmann constant and $T$ to the temperature of the sample. The difference between the populations $N_\alpha$ and $N_\beta$ results in the generation of a macroscopic net magnetic moment along $B_0$ ($z$-axis)[5]. However, since $\Delta E << k_B T$ at ambient temperature and at feasible magnetic fields on Earth, the polarization attained is on the order of ppm at clinically applied $B_0$ (1.5 T)

with a field $B_\mu << B_0$, where $B_\mu$ is the field generated by the nuclear magnetic moment ($\mu$)[1].

The implication of this result is that techniques based on nuclear magnetism are generally described as insensitive[6]. Additionally, due to this insensitivity, directly directing $B_\mu$ along $B_0$ with a magnetometer is not generally feasible with limited exceptions, such as solid hydrogen at 2 K[5,7]. Instead, detection of $B_\mu$ follows after perturbing the system, which typically takes the form of a radio frequency pulse[5]. In a nuclear magnetism context, a perturbation refers here to any process that manipulates the direction of the $B_\mu$ field, including radio frequency pulses and alternation of the orientation of $B_0$. $B_\mu$ is typically then detected via nuclear magnetic resonance (NMR); this is most simply achieved by applying a radio frequency pulse at the Larmor frequency ($\nu = \gamma B_0/2\pi$), orthogonal to the $B_0$ axis (a $\pi/2$ pulse along the $xy$-plane), which rotates the orientation of $B_\mu$ from its equilibrium along the $B_0$ axis to the transverse $xy$-plane. As the system returns to equilibrium, $\mu$ precesses around the z-axis, generating a periodic magnetic signal that can be observed by a detector, such as a Faraday coil, oriented to be sensitive perpendicular to $B_0$[8,9].

The rate a system of nuclear spins returns to their thermal equilibrium, aligned along $B_0$, is described by the spin-lattice relaxation time, $T_1$. Since the

[1]Physikalisch-Technische Bundesanstalt (PTB), Berlin, Germany. [2]Institute of Molecular Biology and Medicinal Chemistry, Medical Faculty, Otto-von-Guericke University Magdeburg, Magdeburg, Germany. [3]High-Field Magnetic Resonance Center, Max Planck Institute for Biological Cybernetics, Tübingen, Germany. [4]Section Biomedical Imaging, Molecular Imaging North Competence Center (MOIN CC), Department of Radiology and Neuroradiology, University Hospital Schleswig-Holstein and Kiel University, Kiel, Germany. ✉e-mail: john.myers@ptb.de; rainer.koerber@ptb.de

**Fig. 1 | Polarization transfer for SABRE-SHEATH of [1-$^{13}$C]pyruvate.** Polarization transfer of spin order from pH$_2$ to the $^{13}$C and $^1$H nuclei of [1-$^{13}$C]pyruvate is shown in blue. The process is catalyzed by an iridium-based catalyst (Ir bound to a 1,3-bis(2,4,6-trimethylphenyl)imidazol-2-ylidene (IMes) moiety) that coordinates pH$_2$, [1-$^{13}$C]pyruvate and the co-ligand dimethyl sulfoxide (DMSO). The $J$-coupling constants of the iridium-bound, pH$_2$ hydride, $J^{HH} = -10.5$ Hz, and the $^{13}$C and $^1$H nuclei, $J^{CH} = 1.27$ Hz, are shown.

signal from NMR is proportional to the number of nuclei perturbed to align along the $xy$-plane, a well-known $T_1$ becomes essential for determining the period between radio frequency pulses. This becomes especially relevant in techniques, such as MRI or multi-dimensional NMR, which rely on many sequential acquisitions to perform a complete sequence[1,10]. However, the direct observation of $T_1$-relaxation for understanding nuclear magnetism via NMR is not possible, since the decay of the periodic signal of an NMR experiment is dominated by the faster loss of coherence, described by the spin-spin relaxation constant, $T_2$ ($T_2 \leq T_1$), chemical exchange and magnetic field inhomogeneity[5]. Therefore, $T_1$ is usually determined indirectly, using specialized sequences, such as the classic inversion recovery sequence[11,12].

A disadvantage of this particular technique is that it can be relatively time-consuming. Additionally, it is also unsuitable for transiently polarized systems, since the process behind each acquisition destroys all longitudinal nuclear magnetization. While this can be addressed by techniques, such as performing sequential acquisitions after pulses much less than $\pi/2$ (small flip angle), which can decrease this time to several times $T_1$[13], this imposes the additional procedure of also having to account for the effect of the repeated radio frequency pulses. Additionally, low sensitivity begins to become a problem again, due to the decreased signal from the low flip angle pulses. The direct characterization of $T_1$-relaxation, via unperturbed direct detection, begins to become rather attractive, since in contrast to inversion recovery, the number of points used to determine $T_1$ for the exponential fit is no longer proportional to the number of experiments performed. Instead, it is proportional to the sampling rate of the data acquisition system, provided the SNR is large enough. This feature also permits easier observation of deviations from a mono-exponential fit, which would be missed by the lower temporal resolution of the other aforementioned methodologies. Therefore, the unperturbed direct detection of nuclear magnetism represents an alternative technique for the better understanding of nuclear magnetism by enabling rigorous characterizations of the relaxation of nuclear magnetism systems with unparalleled temporal resolution.

However, there still remains the challenge of sensitivity. At thermal equilibrium, even the most sensitive magnetometers would be unable to detect $B_\mu$ in most cases, including the wide field of nuclear magnetism in solution at ambient or near ambient temperatures (~20°C). One method for addressing this is the use of hyperpolarization techniques. These include methods based on parahydrogen[14], dynamic nuclear polarization[15] and the nuclear Overhauser effect[16], among many others. A hyperpolarization technique of interest is signal amplification by reversible exchange (SABRE)[17]. A strength of this solution-based technique is that it can continuously generate hyperpolarized substrate in situ, allowing for multiple measurements of hyperpolarized signal over a lengthy period (hours). This eliminates the need to prepare multiple samples and minimizes the effect of sample variability over repeated measurements, allowing for easier acquisition of unperturbed direct detection data.

A particular variant of interest is SABRE in shield enables alignment transfer to heteronuclei (SABRE-SHEATH), since it permits the hyperpolarization of heteronuclei, such as $^{15}$N and $^{13}$C, which often have longer spin-lattice relaxation times than $^1$H and are prevalent in molecules of biological interest[18,19]. Using fields on the order of hundreds of nT to some μT, it is possible to polarize these nuclei to the order of percents; this level has been shown in prior dissolution dynamic nuclear polarization (dDNP) studies to

be high enough to enable pre-clinical, human metabolic studies and cancer diagnosis[20–23]. One such molecule of interest for dDNP studies on human metabolism and cancer is $^{13}$C labelled pyruvate. In 2019, Iali et al. were also able to demonstrate its hyperpolarization, using SABRE-SHEATH, making it also a molecule of interest for unperturbed direct detection[24].

In SABRE-SHEATH of [1-$^{13}$C]pyruvate, the pyruvate forms a complex with an iridium-based catalyst and parahydrogen (pH$_2$), a pure state spin isomer of hydrogen[25]. If a magnetic field ($B_{Hyp}$) is applied around the level anti-crossing condition: $B_{LAC} \sim (2\pi J^{HH})/(\gamma^{1H} - \gamma^{13C})$ (Ex. ~ 330 nT for [1-$^{13}$C]pyruvate, given $J^{HH} = -10.5$ Hz[26]), spin order from pH$_2$ can be transferred to the $^1$H and labelled $^{13}$C of the [1-$^{13}$C]pyruvate, leading to the generation of a hyperpolarized spin system along $B_{Hyp}$ (Fig. 1)[22,27]. However, even with hyperpolarization, the generated $B_\mu$ is still typically over 10 000 × smaller than the applied $B_{Hyp}$ field, making it infeasible, for most types of magnetometers to perform unperturbed direct detection[28]. Additionally, there is also the general challenge of detecting a DC signal. The oft-employed Faraday coil for NMR experiments measures the rate of change of magnetic flux, and thus cannot detect DC signals. This can be circumvented through the use of alternative magnetometers, such as a superconducting quantum interference device (SQUID) (Fig. 2a) or an optically pumped magnetometer (OPM). Since these sensors measure flux density rather than the rate of change of magnetic flux, they are able to detect DC signals and are capable of detecting signals on the order of ~1 pT or higher (Fig. 2b, which is about what would be expected from the hyperpolarization of [1-$^{13}$C]pyruvate by SABRE-SHEATH[29]). Early pioneering work by Cohen-Tannoudji[30] has demonstrated the direct detection of the static magnetic field of hyperpolarized $^3$He gas using an OPM. More recent studies focussing on detecting hyperpolarized pyruvate with OPMs, measuring sensitive to the $B_0$ axis, probed $T_1$, by repeatedly inverting $B_\mu$ ($\pi$ pulses)[31,32]. In contrast, this work demonstrates the unperturbed, direct detection of $B_\mu$ at ambient pressure and temperature, in solution for the first time to the best of our knowledge, using a SQUID-based magnetometer and hyperpolarization via SABRE-SHEATH. This allowed us to directly observe hyperpolarization and relaxation of [1-$^{13}$C]pyruvate at fields on the order of hundreds of nT in real-time with unparalleled, temporal resolution, giving insight into how the hyperpolarization process of [1-$^{13}$C]pyruvate via SABRE-SHEATH works as a function of $B_{Hyp}$.

## Results

To determine conditions to probe, for the unperturbed, direct detection, the dependency of the signal originating from the $^{13}$C nucleus of [1-$^{13}$C]pyruvate as a function of $B_{Hyp}$ was characterized with NMR. This was realized with an ultra-low field SQUID-based MRI setup, using non-adiabatic magnetic field cycling within a moderately shielded, magnetically shielded room (MSR).

Unperturbed, direct detection of [1-$^{13}$C]pyruvate was then performed by measuring the magnetic flux with a SQUID, during bubbling of pH$_2$ into the [1-$^{13}$C]pyruvate sample (polarization buildup) without any other system perturbation. The spin-lattice ($T_1$) relaxation was characterized when bubbling was stopped, and the signal began to decay. This was performed at the optimal $B_{Hyp}$ field, maximizing $^{13}$C polarization. The unperturbed direct detection of $B_\mu$ was then confirmed by comparison with an acquisition at a sub-optimal $B_{Hyp}$ field, and with a non-hyperpolarized control, where the pH$_2$ was replaced with nitrogen gas. Time constants for the hyperpolarization and the spin-lattice relaxation were also extracted from first-order exponential fits of the data. For comparison, at the optimal $B_{Hyp}$ field, the time constants for buildup of hyperpolarization and spin-lattice relaxation were also determined, using NMR with the same setup as that used for direct detection.

### SABRE-SHEATH of [1-$^{13}$C]pyruvate via NMR, using non-adiabatic magnetic field cycling

An example NMR spectrum of [1-$^{13}$C]pyruvate hyperpolarized with SABRE-SHEATH is shown in Fig. 3a at a detection field of 38.5 μT. For

**Fig. 2 | System setup and noise performance.**
**a** Ultra-low noise direct detection setup. The "SABRE reactor" and direct current SQUID detector are located inside an ultralow field MRI setup within a moderately magnetically shielded room ("ZUSE MSR"). The $B_{Hyp}$ field is generated by the $z$ Helmholtz coil, and for field cycling experiments, the detection field, $B_{Det}$ is generated by the $x$ Helmholtz coil. The sample region of the SABRE-SHEATH reactor lies underneath a pick-up coil operated as a first-order axial gradiometer, coupled inductively to a direct current SQUID. The SABRE reactor and SQUID dewar are tilted by 5°, as shown in the figure, to aid backflow of sample upon bubbling.
**b** Characterization of the noise performance of the direct detection setup. Both with $B_{Hyp}$ on and off, a white noise performance of ~ 200 aT/$\sqrt{Hz}$ is attained, and in the cutaway, field variation is < 0.7 pT.

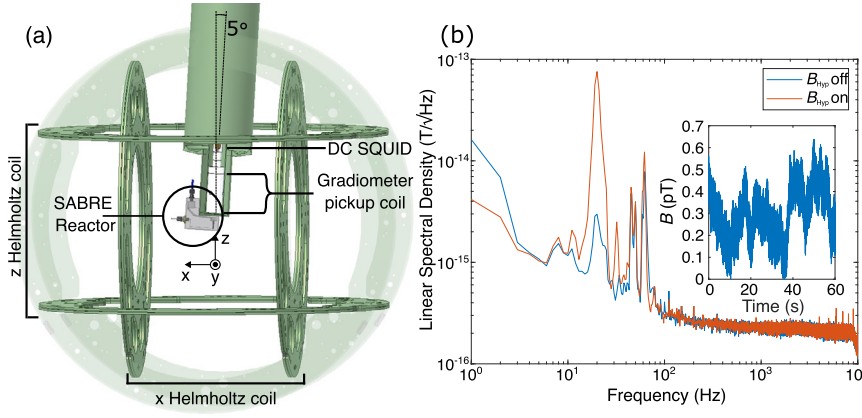

[1-$^{13}$C]pyruvate, both the $^{13}$C and $^1$H signals undergo splitting into a quadruplet and a doublet, respectively. This is due to the $J$-coupling, measured to be 1.27 Hz, between the $^{13}$C and $^1$H nuclei of the [1-$^{13}$C]pyruvate molecule, where in prior work, the negative $^1$H signal is attributed to dissolved, hyperpolarized orthohydrogen[33]. The magnitude of $B_\mu$ is primarily dominated by the magnetization of the $^{13}$C nuclei; the signal at the time of field cycling, attributable to the $^{13}$C nucleus of [1-$^{13}$C]pyruvate plotted against $B_{Hyp}$ is shown in Fig. 3b. The largest signal was achieved at a $B_{Hyp}$ of 500 nT, where the $^{13}$C signal was determined to be 6.20 ± 0.34 pT.

### Unperturbed direct detection of nuclear magnetism from [1-$^{13}$C]pyruvate hyperpolarized by SABRE-SHEATH

The hyperpolarization of [1-$^{13}$C]pyruvate via SABRE-SHEATH and its relaxation were observed at the optimal $B_{Hyp}$ of 500 nT. The results of five consecutive six minute cycles of hyperpolarization (3 minutes) and relaxation (3 minutes) are shown in Fig. 4. Over the six cycles, a total change in field of 9.68 ± 0.94 pT was observed, corresponding to a sample polarization of 0.509% ± 0.049% (for reference, for the setup applied here, given a typical pre-clinical sample with 10% polarization of 70 mM [1-$^{13}$C]pyruvate, we would have detected a $B_\mu$ of 217 pT). At the start of each hyperpolarization phase, a slight decrease in $B$ is observed, whereas the reverse is true at the start of each relaxation phase.

For the sub-optimal acquisition, a $B_{Hyp}$ field of 150 nT was chosen in particular to maximize any potentially observed, field-dependent effects, while still being able to characterize both hyperpolarization and relaxation at the $B_{Hyp}$ field. As shown in Fig. 3b, at a $B_{Hyp}$ of 100 nT, the lowest $B_{Hyp}$ field characterized, an 81% reduction in signal is observed versus at the optimal $B_{Hyp}$ of 500 nT. The slightly higher sub-optimal $B_{Hyp}$ field of 150 nT was chosen for unperturbed direct detection, since the observed 1.20 ± 0.05 pT at a $B_{Hyp}$ of 100 nT was close to our system's detection limit (~0.6 pT at near DC). This acquisition at a $B_{Hyp}$ of 150 nT was then performed in conjunction with a second trial at a $B_{Hyp}$ of 500 nT and a nitrogen control. As shown in Fig. 5a, the same features observed at a $B_{Hyp}$ of 500 nT are also observed at the sub-optimal $B_{Hyp}$ of 150 nT. However, the total amplitude of the signal at a $B_{Hyp}$ of 150 nT is ~60% smaller than observed at the optimal $B_{Hyp}$ of 500 nT. This is in contrast to the control experiment, where nitrogen gas instead of pH$_2$ was bubbled at the optimal $B_{Hyp}$ field of 500 nT. In this case, no significant change in flux was observed.

The first-order exponential fits, where the hyperpolarization buildup ($T_B$) and spin-lattice relaxation ($T_1$) times were extracted from are shown in Fig. 5a. The $T_B$ and $T_1$ for both [1-$^{13}$C]pyruvate samples at the optimal 500 nT $B_{Hyp}$ field and for the second sample at the sub-optimal 150 nT $B_{Hyp}$ are shown in Table 1. Between the two runs (S1 and S2), performed on different days, at the optimal 500 nT $B_{Hyp}$ field, the time constants, $T_B$ and $T_1$, differed from one another by less than 2 s, a < 4% difference. Also of note

is that the sub-optimal 150 nT $B_{Hyp}$ data set had the shortest $T_1$ in addition to a $T_B$, nearly 40% larger than the $T_B$ at the optimal 500 nT $B_{Hyp}$ field.

### Buildup time and spin-lattice relaxation times via NMR

The $T_B$ and $T_1$ of [1-$^{13}$C]pyruvate hyperpolarized via SABRE-SHEATH were also determined at the optimal $B_{Hyp}$ field via NMR on a separate sample to the direct detection ones. The buildup of hyperpolarization was characterized by varying the total polarization time, while the spin-lattice relaxation was characterized by introducing a delay between polarization and the beginning of an NMR acquisition. The result of the buildup characterization is shown in Fig. 5b. Due to the large role that the final point of an exponential fit plays in determining the time constant, the fit for $T_B$ was performed both over all the data and over all the data except at a polarization time of 180 s. For these two fits, $T_B$ was determined to be 37.0 s with a 95% confidence interval (CI) of [33.5 s, 41.4 s] versus 26.0 s 95% CI [21.0 s, 34.2 s], respectively. The result of the spin-lattice characterization via NMR is shown in Fig. 5c, where the summed amplitude of the $^{13}$C signals from [1-$^{13}$C]pyruvate hyperpolarized via SABRE-SHEATH is plotted against the delay time between the cessation of polarization and the start of the NMR acquisition. Of note, the amplitude of the $^{13}$C signals initially increases as the time between cessation of bubbling of pH$_2$ gas and start of the NMR acquisition increases, up to a delay of 10 s. This signal increase is a factor two times higher for a delay of 10 s versus performing an NMR acquisition immediately after cessation of bubbling pH$_2$ gas. $T_1$ was extracted from a first-order exponential fit over the data excluding the period where the signal continued to increase after cessation of bubbling (the first ten seconds) and was determined to be 43.3 s 95% CI [38.2 s, 50.2 s].

### Discussion
#### Direct detection
The results from the Section, "Unperturbed direct detection of nuclear magnetism from [1-$^{13}$C]pyruvate hyperpolarized by SABRE-SHEATH" demonstrate that unperturbed direct detection of $B_\mu$ in solution is possible, upon sample hyperpolarization and application of an ultrasensitive detection setup. When pH$_2$ was being bubbled into the sample, the detected field reached 9.68 ± 0.94 pT, decaying to zero when bubbling was stopped. In contrast, when the pH$_2$ is replaced with nitrogen gas, no significant changes in the detected field are observed.

In prior work from Eills et al.[31], detection of longitudinal magnetization was demonstrated using an OPM-based system. However, to compensate for magnetic field drifts and fluctuations, they used a train of repeated $\pi$ pulses with signal recording and averaging between pulses. In contrast, we did not find it necessary to apply these repeated $\pi$ pulses to compensate for magnetic field drifts from the environment, especially since doing so would have compromised our temporal resolution. Instead, we demonstrated that

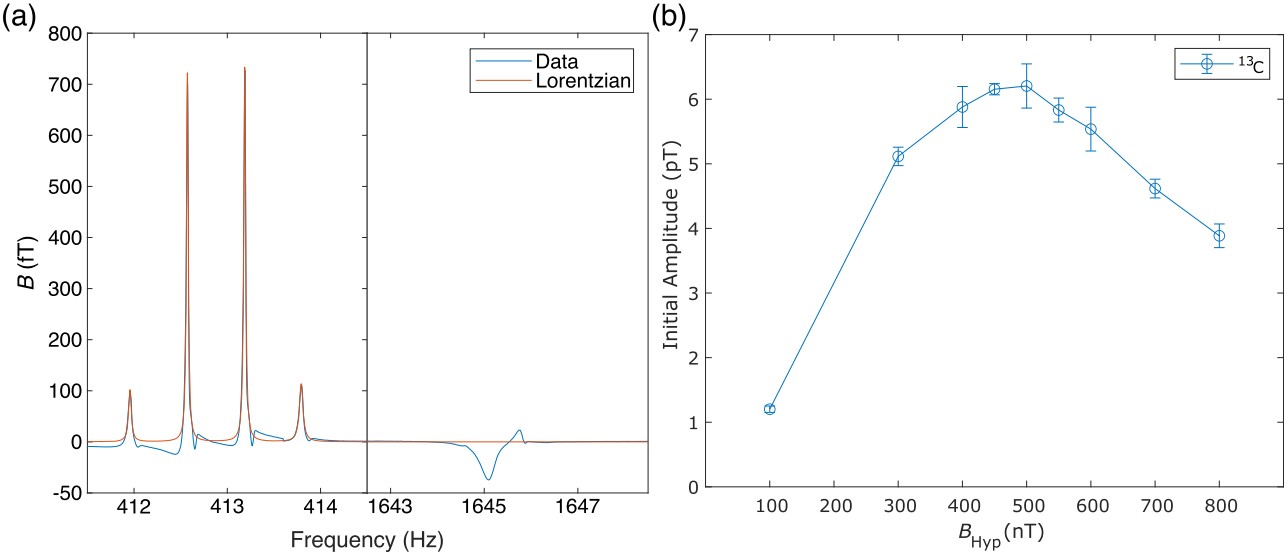

**Fig. 3 | Example NMR spectrum of [1-¹³C]pyruvate and dependency of amplitude on $B_{Hyp}$. a** NMR spectrum of [1-¹³C]pyruvate collected at a $B_{Hyp}$ of 500 nT and a $B_{Det}$ of 38.5 μT with the Lorentzian fits for the ¹³C signals plotted on top. **b** Summed initial amplitudes of the four ¹³C signals, determined from their Lorentzian fits versus $B_{Hyp}$ ($N = 5$).

**Fig. 4 | Unperturbed direct detection of nuclear magnetism from [1-¹³C]pyruvate.** Direct detection of the buildup of polarization (pH₂ in) from SABRE-SHEATH and subsequent spin-lattice relaxation (No pH₂) of [1-¹³C]pyruvate at a $B_{Hyp}$ of 500 nT. (Sample 1).

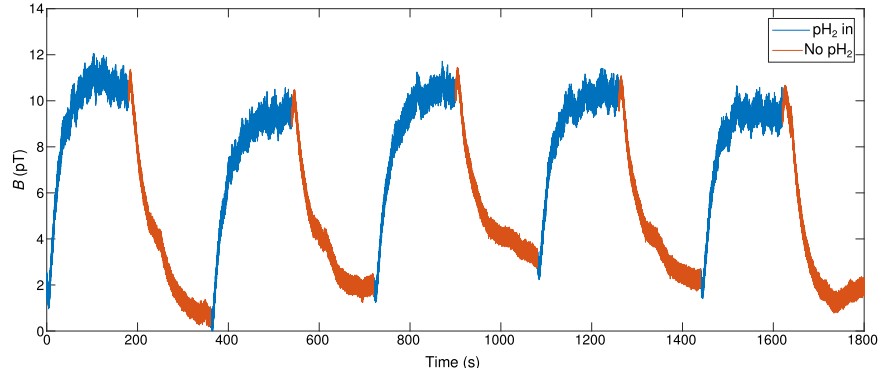

our use of a gradiometer SQUID detection setup (see the "Experimental setup" section) was sufficient to compensate for this to allow for continuous detection at a sampling rate up to the limits of our read out setup (20 kHz for these measurements); this enables essentially real-time observation of polarization build-up and decay in contrast to conventional NMR, where typically 10–20 points are measured.

### $T_B$ and $T_1$ at ultra-low field

Due to time constraints, in our characterization of $T_B$ and $T_1$ of [1-¹³C]pyruvate hyperpolarized via SABRE-SHEATH by NMR, we were only able to acquire seven points for the buildup of polarization and five points for the spin-lattice relaxation, where each point was acquired five times to allow for statistics. Doing so required approximately six hours of experimental time, during which the last hour started to see significant sample loss, due to evaporation into the environment (see SI Fig. S1). It is this relative lack of points for the exponential fit that led to the fairly large 95% confidence intervals of the exponential fits (up to $\pm 8$ s for $T_B$ and up to $\pm 7$ s for $T_1$). In contrast, the 95% confidence intervals of the fits for the direct detection data are not reported, due to being less than $\pm 0.1$ s, which is a result of the orders of magnitude more points available for the fit.

Comparing the NMR data with the direct detection data, we find for the characterization of $T_B$, there appears to be a possible discrepancy between the ~26 s determined via direct detection versus the 37 s

determined via NMR, even accounting for the uncertainty of the fit for the NMR data. One possible explanation for this is the outsized influence of the final point of the exponential fit (buildup time 180 s). This is demonstrated that by its exclusion here, the $T_B$ from the fit decreases from 37 s to the 26 s also observed via direct detection. Regarding possible reasons why the final point of the buildup may have been unreliable is that it was the final point acquired, during the NMR characterization, which was performed as the sample within the sensitive area for our detector began to rapidly evaporate. This could have impacted the reliability of the methodology we used to account for sample loss (see SI Supplementary Note 1). For the characterization of the spin-lattice relaxation, both our NMR characterization and direct detection characterization of hyperpolarized [1-¹³C]pyruvate at 500 nT are in agreement with one another, evidenced by both methodologies yielding a $T_1$ of ~43 s.

A $T_1$ of 43 s at 500 nT is also consistent with prior work from Chattergoon et al., where the spin-lattice relaxation of an aqueous solution of [1-¹³C]pyruvate was characterized, using NMR from 237 μT to 0.705 T[13]. Upon extrapolation of their results, not only is our $T_1$ of ~43 s at a $B_{Hyp}$ of 500 nT consistent with their results, but also our $T_1$ of ~40 s at a $B_{Hyp}$ of 150 nT. The validity of using unperturbed direct detection is additionally supported by the less than 4% variability in measured $T_1$ between the two independently prepared [1-¹³C]pyruvate samples. More recently, the $T_1$ relaxation of a solution of [1-¹³C]pyruvate with trityl radical after dissolution

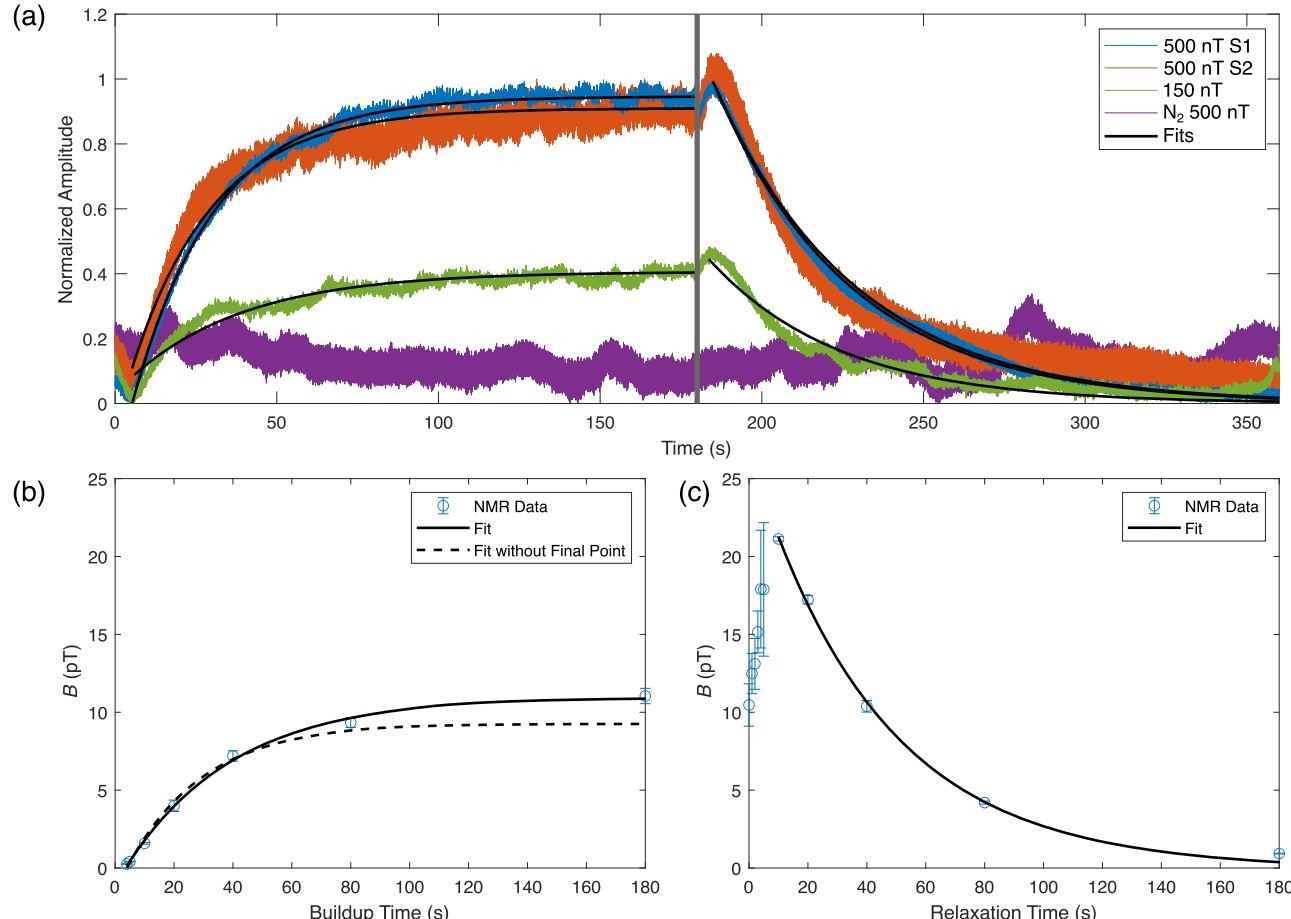

**Fig. 5 | Characterization of polarization buildup and spin-lattice relaxation by direct detection and NMR. a** Average buildup-relaxation cycle from the direct detection of [1-$^{13}$C]pyruvate hyperpolarized with SABRE-SHEATH at multiple $B_{Hyp}$ fields ($N = 2$ for $B_{Hyp}$ 150 nT (S2), otherwise $N = 5$). Data is normalized to the maximum of the polarization buildup phase at the optimal $B_{Hyp}$ field of 500 nT. "S1" and "S2" refer to data collected from independently prepared samples. First-order exponential functions were fitted for the buildup and relaxation phases, except for the nitrogen control, where pH$_2$ was replaced by nitrogen. **b** Buildup time ($T_B$) as measured via NMR. The x-axis corresponds to the time pH$_2$ was bubbled into the SABRE reactor prior to NMR acquisition. The y-axis shows the summed initial

amplitude of the four $^{13}$C signals, determined from their Lorentzian fits, as in Fig. 2b, corrected for sample loss. For each point $N = 5$. A first-order exponential function was fitted over all the data, and a separate first-order exponential function was fitted over all the data except the acquisitions performed with a bubbling time of 180 s. **c** Spin-lattice relaxation ($T_1$) as measured via NMR. The x-axis corresponds to the delay between stopping the bubbling of pH$_2$ into the SABRE reactor and performing field cycling to begin the NMR acquisition. All other features are analogous to Fig. 5b ($N = 5$). The first-order exponential fit was performed over all data from a relaxation time of 10 s onwards. Prior points were disregarded to account for fluid dynamic effects affecting sample polarization.

### Table 1 | $T_B$ and $T_1$ for [1-$^{13}$C]pyruvate hyperpolarized via SABRE-SHEATH

| Sample | $T_B$ (s) | $T_1$ (s) |
|---|---|---|
| 500 nT S1 | 25.7 | 42.0 |
| 500 nT S2 | 26.0 | 43.6 |
| 150 nT S2 | 35.9 | 40.4 |

dynamic polarization was characterized from 7.8 μT to 9.4 T[34]. At ultra-low field, they measured a $T_1$ between 40 and 52 s, which is also consistent with our results. In their case, the high variability in $T_1$ was attributed to paramagnetic impurities[34].

### Influence of sample variation and gas-fluid dynamics on hyperpolarization

One feature of note in Fig. 5a is that the data from S2 appear noisier than the data from S1, and that the data acquired at a $B_{Hyp}$ of 500 nT appear noisier than the data acquired at a $B_{Hyp}$ of 150 nT. We attribute the latter feature primarily to the vibrational noise caused by operating the coil, used to generate the $B_{Hyp}$ field. This is most strongly evidenced by the difference in

noise observed between the two S2 samples. Since 500 nT is about 3.3 times smaller than 150 nT, the amplitude of vibrational noise originating from operating a coil at 500 nT versus 150 nT would also be expected to be 3.3 times greater, which is the case here (see Fig. S3). Regarding the differences between S1 and S2, as stated before, the data acquired at a $B_{Hyp}$ of 500 nT was normalized to allow for comparing the two samples. This was done, due to sample variability, which we attribute to variability in the pyruvate sample, the SABRE reactor and the positioning of the SABRE reactor within the system. We propose that a combination of these three factors led to both a lower absolute field for S2 versus S1, in addition to increased vibrational noise between S2 and S1.

Perhaps the most unexpected feature from measuring the polarization buildup and decay was, however, an increase in $B$ at the beginning of each relaxation phase (no bubbling) and the decrease in $B$ at the beginning of each hyperpolarization phase (bubbling), where the former was also observed in our NMR characterization of $T_1$. These characteristics cannot be explained by SABRE-SHEATH hyperpolarization and spin-lattice relaxation alone. However, one possible explanation is that the field detected varied at these points, due to variable sample distance from the SQUID detector, caused by sample displacement from the flowing pH$_2$ gas. At the beginning of each spin-lattice relaxation phase, the cessation of flow of the pH$_2$ gas into the

**Fig. 6 | Possible polarization complexes for hyperpolarization of [1-$^{13}$C]pyruvate via SABRE-SHEATH.** Ir-IMes catalyst [1-$^{13}$C]pyruvate that can lead to $^{13}$C polarization via SABRE-SHEATH. Complex I shows how [1-$^{13}$C]pyruvate may bind equatorially, while Complex II shows how [1-$^{13}$C]pyruvate may bind axially-equatorially.

sample could lead to this space being taken up by previously displaced [1-$^{13}$C]pyruvate sample, decreasing the distance between the sample and the detector, thereby increasing the observed flux. We also considered the effect from the two-compartment sample chamber geometry we used for our experiments and any possible effects from this in prior work (see Myers et al. for details on sample geometry and effects[29]). However, our simulations indicate that the signal from the second compartment is negligible and could not have caused this ~10% increase in field after pH$_2$ supply was started or stopped, during direct detection, nor the ~ 100% increase in field observed with NMR. In follow-up experiments with an alternative reactor design (see ref. 35), this behaviour was reproducible, but not quantitatively, suggesting a fluid dynamics cause. The different magnitudes of this effect seen by direct detection versus NMR are an area of interest for future work. Regarding the decrease in signal at the beginning of each hyperpolarization phase, this likely would have otherwise been missed, without the very rapid observation of magnetization evolution, which direct detection here enabled. Considering that this is caused by the gas-fluid dynamics, one could also use direct detection for an analysis of the performance of SABRE reactors and use it to tune the reactor geometry for optimized SABRE polarization.

### Effect of magnetic field on polarization

Using the results of unperturbed direct detection on [1-$^{13}$C]pyruvate hyperpolarized via SABRE-SHEATH at the sub-optimal $B_{Hyp}$ field of 150 nT versus the optimal $B_{Hyp}$ field of 500 nT, it was additionally possible to characterize how [1-$^{13}$C]pyruvate hyperpolarizes via SABRE-SHEATH as a function of the $B_{Hyp}$ field. As shown in Fig. 5a and Table 1, both the maximum polarization attained and the $T_B$ show dependency on the $B_{Hyp}$ field. A compelling explanation for this observation is that [1-$^{13}$C]pyruvate hyperpolarizes in two different complexes with the Ir-IMes catalyst (Fig. 6), as has been proposed in prior work[36]. For the aforementioned $B_{LAC}$ value of 330 nT, [1-$^{13}$C]pyruvate is coordinated equatorially with the Ir-IMes catalyst (Complex I). As a result, the $J$ coupling between the two hydrides of the pH$_2$ is ~ −10.5 Hz, resulting in the previously quoted field at the level anticrossing, $B_{LAC}$, of 330 nT, where maximum polarization would be expected. Due to this work being performed at ambient temperature (22° C), the optimum spin order transfer (SOT) field is elevated to the observed ~500 nT $B_{Hyp}$, due to chemical exchange, where the complex dissociation rate constant $k_d$ is between 85-115 s$^{-1}$[36]. However, it is also possible for [1-$^{13}$C]pyruvate to coordinate axially with the Ir-IMes (Complex II)[24], although less favourably than in the equatorial complex. In this complex, the $J$ coupling between the pH$_2$ hydrides is ~ −6.5 Hz, instead of the prior −10.5 Hz, resulting in a lower $B_{LAC}$ of 200 nT for Complex II. Additionally, the dissociation constant for Complex II is significantly lower than that for Complex I, where $k_d$ is between 20–25 s$^{-1}$[36]. The implication of this is that the optimum SOT-field for Complex II is less elevated for its $B_{LAC}$ at ambient temperature versus for Complex I, resulting in a slower SOT rate for Complex II versus Complex I.

Assuming that both Complexes I and II can lead to hyperpolarization of [1-$^{13}$C]pyruvate via SABRE-SHEATH, then the sub-optimal $B_{Hyp}$ field of 150 nT would be closer to the optimal SOT-field for Complex II than to the

optimal SOT-field for Complex I. As a result, [1-$^{13}$C]pyruvate would hyperpolarize primarily via Complex II at this lower field. Due to the slower SOT and lower favourability of Complex II versus Complex I, this would then lead to a longer $T_B$ and lower signal. Both of these features are observed upon the unperturbed direct detection of the hyperpolarization of [1-$^{13}$C]pyruvate via SABRE-SHEATH at the $B_{Hyp}$ field of 150 nT, highlighting the need for ultra-low field magnetization sensing techniques.

### Coherent transfer of polarization

While not observed in this proof-of-concept, using unperturbed direct detection, it should also be possible to observe predicted oscillations in polarization, during SABRE-SHEATH, where coherent polarization transfer in a transient complex is a mechanism of SABRE polarization[37,38]. However, even when polarization production is continuous, due to rapid chemical exchange and restarting of polarization transfer in the complex, the characteristic oscillations could be suppressed, yielding the observed exponential build up[39–41], unless specialized sequences are used, such as SABRE with alternating magnetic fields[42–45] or the recently proposed photo-triggered SABRE[46].

In future work, we will apply direct detection to such methods to unlock observation of real-time coherent polarization transfer, which is critical for further fine-tuning of hyperpolarization and for a better understanding of multi-parametric dynamics, including spin evolution and chemical exchange. Understanding these effects and being able to quickly characterize hyperpolarization and spin-lattice relaxation are necessary components for the optimization of hyperpolarization probes for future in vivo metabolic studies and clinical applications.

## Methods

### [1-$^{13}$C]pyruvate SABRE-SHEATH sample preparation

An 8 mL sample was made up to the following final concentrations in methanol (MeOH) (Roth, Rotisolv ≥99.9% 67-56-1): 50 mM [1-$^{13}$C]pyruvate (sodium pyruvate-1-$^{13}$C, Sigma-Aldrich 490709), 5 mM IrIMes(COD) complex (synthesized, as described in ref. 47), 18 mM dimethyl sulfoxide (DMSO). The pyruvate salt and IrIMes catalyst were dissolved via sonication, and prior to experiments, samples were degassed with sonication.

Nominally 99% parahydrogen (pH$_2$) was generated in a home-built, liquid-helium dipstick. The dipstick was equipped with a platinum resistance thermometer, enabling temperature regulation via proportional-integral (PI) control. Ortho to para conversion was catalyzed at 25 K via the use of a ferric oxide catalyst.

### Experimental setup

The experiments were performed in a moderately shielded, magnetically shielded room, "ZUSE-MSR"[48]. Inside the MSR, the reactor was placed such that the main sample area was placed directly underneath the detecting element. The detecting element was a first-order gradiometer pick-up loop (45 mm diameter, 120 mm baseline), sensitive along the z-axis, coupled inductively to a low T$_c$ current sensing DC-SQUID. The pick-up loop and the SQUID were both housed in the ultra-low noise liquid helium dewar, LINOD2[49], leading to a cold-to-warm distance of 18 mm from the top of the sample to the bottom of the pick-up loop. The dewar, including the reactor, was tilted 5° off the z-axis to enable better backflow in the reactor to the sample area; the sample area of the reactor was centred within the already existing ultralow field MRI setup within the room[50]. See Fig. 2a for more details (see SI Fig. S2 for a picture).

The noise performance of the system was characterized by performing a 60 s acquisition of the setup at 20 kHz (1,200,000 points). The data were then separated into one-second intervals that were each individually detrended (linear fit along the interval). The average linear spectral density was then plotted (Fig. 2b).

Prior to experiments, the iridium complex was activated by bubbling pH$_2$ into the reactor at 1 L/h at ambient temperature (22° C) and pressure for half an hour.

**Field cycling**. Field cycling experiments for characterizing $B_{\text{Hyp}}$ were performed by bubbling pH$_2$ for 40 s at 2 L/h at ambient temperature and pressure into the sample, exposed to the $B_{\text{Hyp}}$ field. Then, the $B_{\text{Hyp}}$ field was non-adiabatically switched off and a 38.5 µT detection field, $B_{\text{Det}}$, orthogonal to $B_{\text{Hyp}}$, was non-adiabatically switched on. Upon field switching, pH$_2$ also shunted away from the reactor via the use of a three-way valve to minimize noise from sample turbulence. Acquisition was then performed for 40 s at 20 kHz (800 000 points).

Field cycling experiments for characterizing $T_{\text{B}}$ and $T_1$ were performed similarly. For the characterization of $T_{\text{B}}$, the bubbling time was varied from 4–180 s. Acquisition then proceeded exactly as performed to characterize $B_{\text{Hyp}}$. For the characterization of $T_1$, pH$_2$ was bubbled for 180 s at 2 L/h at ambient temperature and pressure into the sample to ensure maximum polarization. Afterwards, the pH$_2$ was shunted away from the reactor. Field cycling and acquisition then proceeded after a delay of 0–180 s, where the field cycling and acquisition were performed as for the characterization of $B_{\text{Hyp}}$.

**Direct detection**. For direct detection, no field switching was performed, and the $B_{\text{Hyp}}$ field remained constant over the entire experiment ( ~30 min.). Acquisition was performed at 20 kHz over the entire experimental time, and pH$_2$ was bubbled into the system on a 50% 6 min. duty cycle (3 min. polarization buildup (pH$_2$ in), 3 min. spin-lattice relaxation (pH$_2$ shunted)) at 2 L/h. For S1 and S2, independently constructed SABRE reactors were used that were made to the same geometry.

## Data post-processing

All analyses in the subsequent sections were performed with scripts written in MATLAB.

**Field cycling**. The data were baseline corrected in the time domain to account for magnetic field drift within the ZUSE MSR by demeaning the data and subtracting a fitted one-term polynomial.

Each $^{13}$C signal was phase corrected individually. Since the data were not acquired in quadrature, this was first performed by generating the analytic signal, using the Hilbert transform. Zeroth-order phase correction was then applied to the time domain. The frequency domain was determined by performing a fast Fourier transform (FFT) normalized to sample length (800,000 points) to the data, whose length was zero-padded to the third closest higher power of two (for robustness of the later Lorentzian fit).

Each $^{13}$C signal was assumed to be described by a Lorentzian function:

$$f\left(v; v_0, \alpha\right) = \frac{C}{\pi\alpha\left[1 + \left(\frac{v - v_0}{\alpha}\right)^2\right]} \quad (1)$$

where $C$ is a multiplicative factor to adjust for signal size, $v_0$, the location of the signal centre and $\alpha$, the half-width at half-maximum (HWHM) of the signal. For the zeroth-order phase corrected spectrum, a Lorentzian function was fitted to the relevant [1-$^{13}$C]pyruvate signal, using the trust region reflective algorithm. For a Lorentzian function, the amplitude is given by: $C/(\pi\alpha)$. However, the calculation of the initial amplitude (beginning of acquisition) of a decaying signal also depends on the acquisition length, since the decay of a periodic signal acts as a window function on the data (as opposed to the standard rectangle window). Since the FFT data were already normalized to the sample length, this correction can be applied by dividing by the average of the effective window function (as opposed to its sum). In this case, this function is a normalized decaying mono-exponential, where the decay rate is dependent on the HWHM of the signal (see Eq. (2)),

$$A = \frac{C}{\pi\alpha \times \exp(-2\pi\alpha t)} \quad (2)$$

where $A$ is the initial amplitude of the $^{13}$C [1-$^{13}$C]pyruvate signal of interest and $t$ is the array of time points (the 800,000 points) over which the data

were acquired. Final sample polarization was calculated by discretizing the sample volume of the SABRE reactor into voxels and then using the principle of reciprocity to determine the total magnetic flux seen by the setup. The full procedure is exactly as described in ref. 29.

**Characterization of $T_{\text{B}}$ and $T_1$ via NMR**. Since the characterization of $T_{\text{B}}$ and $T_1$ via NMR required about six hours of experimental time versus the one required to characterize the polarization as a function of $B_{\text{Hyp}}$, the data were corrected for sample loss from evaporation into the environment. This was achieved by regular performance of control measurements, where pH$_2$ was bubbled for 40 s and field cycling and acquisition followed immediately thereafter, as described in the Section, "Field cycling". The summed amplitude of the $^{13}$C signals from [1-$^{13}$C]pyruvate for the control measurements was tracked versus the time they were performed. These amplitudes were then normalized to the maximum of the control measurements. The $^{13}$C signals from the $T_{\text{B}}$ and $T_1$ acquisitions were then corrected for sample loss by dividing by the proportion of the maximum signal from a control measurement for their times of acquisition. This proportion to divide by was determined by linear interpolation between the control measurements (for more details, see SI Supplementary Note 1). $T_{\text{B}}$ and $T_1$ were then determined by fitting first-order exponential functions to the summed $^{13}$C amplitude data corrected for sample loss. For the spin-lattice relaxation time, the fit was performed ignoring the first 10 s of delay between cessation of bubbling and acquisition to negate fluid dynamic effects from the fit. The reported $T_{\text{B}}$ and $T_1$ relaxation times are from the exponential fits, where the uncertainty reported corresponds to the 95% confidence interval of the fit.

**Direct detection**. For optimal noise performance, acquisitions were performed between 01:30 and 04:00, during the service pause of the Berlin Underground to avoid acquisition of magnetic signals originating from it.

The drift within the ZUSE MSR was accounted for by first subtracting out the mean over the entire acquisition period of the data. The data were then separated into sets based on whether the collection was during the buildup or relaxation phases (ex., 30 min. acquisition becomes ten 3 min. alternating sets of buildup and relaxation phases). Then, the midpoint of the range of each of these sets was plotted against time (for no drift, the midpoints should be zero). The magnetic field drift was then accounted for by subtracting a fitted polynomial function to the set range midpoints from the data (2 terms for $B_{\text{Hyp}} = 500$ nT, 1 term for $B_{\text{Hyp}} = 150$ nT and no fit for the N$_2$ control).

Any lingering effects of magnetic field drift were removed by individually demeaning each complete buildup-relaxation cycle prior to averaging. $T_1$ and $T_{\text{B}}$ were calculated by fitting first-order exponential functions to the averaged buildup and relaxation data sets, removing the first 5 s from the fitting to account for time to allow the system to settle after starting or ceasing flow of pH$_2$. In this case, the 95% confidence intervals are not used as a measure of uncertainty for $T_1$ and $T_{\text{B}}$, since their ranges correspond to less than one tenth of second, which was the precision reported for $T_{\text{B}}$ and $T_1$.

## Presentation of data

The frequency spectrum of the averaged data ($N = 5$) for the data acquired at $B_{\text{Hyp}} = 500$ nT and $B_{\text{Det}} = 38.5$ µT is shown with both first and zeroth order phase correction in Fig. 2a for aesthetic reasons. The first-order phase correction was applied after the FFT step in the handling of the traditional NMR dating by multiplying the result of the FFT by: $\exp\left(-i2\pi t_{\text{Dead}}v\right)$, where $t_{\text{Dead}}$ is the deadtime between non-adiabatic field switching and acquisition (2.619 ms) for the system and $v$ is the frequency corresponding to each point of the double-sided FFT. The result of the Lorentzian fits for the $^{13}$C signals from [1-$^{13}$C]pyruvate are plotted on top.

The dependency of the initial amplitude of the $^{13}$C signals from [1-$^{13}$C] pyruvate versus $B_{\text{Hyp}}$ is shown by plotting the summation of the initial amplitudes of the four $^{13}$C signals versus $B_{\text{Hyp}}$ (Fig. 2b).

For a single 30 min. acquisition at $B_{\text{Hyp}} = 500$ nT, the result after baseline correction, but before demeaning of individual buildup-relaxation

**Article**

cycles is plotted (Fig. 4). To better show the observed exponential behaviour, the averages of the buildups and relaxations are plotted with their exponential fits (Fig. 5a). To account for differences in total signal as a result of originating from different samples, the data were normalized to the maximum of the average buildup data collected at the optimal $B_{Hyp}$ for the relevant sample. The average of the buildup phases is plotted against time: 0–180 s, and the average of the relaxation phases is plotted against time: 180–360 s.

## Data availability

All relevant data are available from the authors upon request to either of the corresponding authors.

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

## Acknowledgements

This project was funded by the DFG (Grant number: 469366436). A.N.P. also acknowledges funding from the German Federal Ministry of Education and Research (BMBF) within the framework of the e:Med research and funding concept (01ZX1915C), DFG (527469039). MOIN CC was founded with a grant from the European Regional Development Fund (ERDF) and the Zukunftsprogramm Wirtschaft of Schleswig-Holstein (Project no. 122-09-053). K.B. acknowledges funding from the DFG (527345502).

## Author contributions

J.Z.M., R.K. performed all experiments at the PTB and data analysis. M.P. synthesized the IrIMes catalyst. K.B., A.N.P., J.Z.M., R.K. and M.P. contributed to the discussion and preparation of the manuscript and approved the final version of the manuscript.

## Funding

## Competing interests

The authors declare no competing interests.
