## [Transparent Peer Review file · Communications Chemistry]

Direct Detection of SABRE-SHEATH Hyperpolarization and Spin-Lattice Relaxation of [1-¹³C]Pyruvate

Corresponding Author: Mr John Myers

Version 0:

Reviewer comments:

Reviewer #1

(Remarks to the Author)

The paper by Myers et al presents a new non-destructive (direct) way of measuring hyperpolarization dynamic using a SQUID. To prove their technique they hyperpolarize [1-¹³C]pyruvate using SABRE-SHEATH. The communication is well-written, and this technique might be useful to characterize spin dynamics phenomena with low SNR and that happen fast with respect to the SNR/time resolution that traditional NMR setup can offer. Nevertheless, I have 2 key concerns the authors should address before recommending this paper for publication.

1- the authors have at their disposal a NMR setup that was used to find the B_{hyp} giving the best signal. I would have expected a measurement of T₁ and T_b using NMR for direct comparison with the new method. Why didn't they do so? I don't think time and SNR is a problem, especially for the hyperpolarized T_b. They cite reference 31, but this is not exactly in the same conditions.

2- in the introduction, the authors point out the limitations of measuring T₁ and T_b of HP signals using NMR and small flip-angles. Although disentangling the effect of RF pulsing from T₁ is not as hard as the authors say (please tone down the sentence at line 144), they have a point. I am not a SQUID expert, but signal intensity is affected by the rate at which the magnetization builds up. After all, we induce a magnetic flux. How do the author take this phenomenon into account? In an exponential growth, the signal varies fast at the beginning and almost doesn't at the end. For a relaxation is the opposite. This can be clearly seen in figure 4, where the SNR is much higher at the beginning of the polarization/relaxation. How this can affect the accuracy of the measurement? And if this is not a problem, please explain why and within which limits (i.e. max detectable rate without errors).

below are some minor comments

line 162: please provide name and references of other HP techniques

line 171: does T_{1s} stand for T₁-SABRE? If so please specify

line 172: 100s is a typo

line 176: remove human studies from a sentence where SABRE appears. SABRE, so far, has not been demonstrated for humans. It could be misleading for people not from the field

line 187: for better understanding, it would be appropriate to provide an estimation of B_{mu} in case a preclinical dose (70 mM) of pyruvate is polarized to let's say 10%

Reviewer #2

(Remarks to the Author)

This study demonstrates the use of SQUID detection in ultralow-field NMR to measure ^{13}C polarization levels, buildup times under SABRE-SHEATH hyperpolarization, and subsequent T1 relaxation. While the results align well with existing literature and the non-perturbative detection approach is compelling, the key advantage of this technique lies in its ability to distinguish between different hyperpolarization pathways or interactions, as evidenced by variations in buildup times with exchange dynamics and magnetic field changes. In this context, the authors propose extending this method to other hyperpolarization techniques to reliably probe the underlying mechanisms of parahydrogen-induced polarization. In principle this approach could be valuable for identifying optimal field conditions to maximize polarization levels, ultimately enhancing NMR sensitivity or MRI imaging using pyruvate or other metabolic probes.

Although the level of scientific work, with the hardware development, the repeatable data at zero field, and efforts in noise reduction are remarkable, the manuscript does not convincingly demonstrate a significant advancement. Since optimal field conditions can already be predicted through level anti-crossing simulations or empirically determined via field cycling in existing platforms, such as those developed by Michael Tyler, James Eills, and collaborators using an OPM detector, the added value of the current approach remains unclear, as similar conclusions could likely be drawn using these alternative methods. Therefore, I would not recommend the manuscript for publication in Communications Chemistry.

In parallel here are some suggestions to improve the manuscript :

In the abstract 'Here, we demonstrate the unperturbed, direct detection of nuclear magnetism, using a SQUID-based detector with frequency-independent sensitivity, enabling perturbation-free measurement. To compensate for low sensitivity at ultra-low magnetic fields, $[1-^{13}\text{C}]$ pyruvate was hyperpolarized with the SABRE-SHEATH technique at 150 nT and 500 nT' The abstract could more clearly explain the rationale for using ultralow field, especially for non-specialists. Rephrasing to introduce that SQUID detectors are well-suited for unperturbed detection at low fields would help clarify the motivation.

On Figure 2, figure 2 the 5° tilt angle from B_0 should be explained in the legend, or at least the methods sections should be referred in the legend. The SABRE-SHEATH reactor should be clearly labeled in the figure, using a color scheme accessible to colorblind readers.

In Figure 4, it is unclear why the polarization does not return to the baseline or why the signal intensity varies between cycles. A brief explanation of these variations and a comment on repeatability would strengthen the interpretation.

The software to treat the data and perform the fittings should be indicated in the Method section.

A supplementary data material would be useful:

- The manuscript should include the fitting equations used for build-up and relaxation curves and report uncertainties in the fitted parameters.
- It would have been nice to have a picture of the setup.
- The methodology for polarization quantification should be clearly described, as in the ArXiv preprint.

Very minor suggestions :

Line 104 : 'which is the most common state probed' can be removed

Line 106-108 : please define Boltzmann distribution with the constant k_B and T

Line 121 : please avoid two consecutive parenthesis for smoother reading.

Line 127 : please use 'time' instead of 'constant'

Line 172 and 201 : please write in letter 'hundreds' instead of '100s'

Reviewer #3

(Remarks to the Author)

This work describes the use of a SQUID detector to directly measure the T1 relaxation ^{13}C pyruvate hyperpolarized using SABRE. It is an elegant combination of two methods to allow for a novel view of polarization buildup and relaxation. The work's scope is limited to a proof-of-concept study but still of interest to the NMR community. The manuscript is well written, and the conclusions are well supported by the results.

Major concerns

I have no major concerns with this work

Minor concerns

1. Between figure 4 and 5 the authors normalize repeated T1 decay measurements to the maximum polarization achieved

during the SABER polarization. However, T1 decay seems to converge to different values for different repetitions as seen in figure 4. Was there some additional data processing done to normalize the lower end of the T1 decay between figures 4 and 5?

2. The exponential fit does not qualitatively appear to perform well on the 2nd 500 nT polarization S2. The fit underestimates the value initially and overestimates it towards the end. Can the authors comment on why this fit appears to have performed worse compared to the first run?

3. The authors need to comment on how the increase in B presumably from geometric sample changes related to gas bubbling were managed during exponential fitting. Where those values not included in the fits, was the initial value for the fit greater than 1?

Version 1:

Reviewer comments:

Reviewer #1

(Remarks to the Author)

The authors have addressed all my comments and I am satisfied by the revised version of the paper. I am happy to suggest this revised version for publication.

Reviewer #2

(Remarks to the Author)

The revised manuscript provided by the authors have addressed most comments from the reviewers well. I would have three minor comments left before publication :

Line 37 : limits -> limit

Line 195 : oft or often ?

Lines 338, 354, 499, 595: please refer to the dedicated section or figure in the SI. Example for line 499, please refer to Figure or Section S1 in the SI instead of 'see SI for a picture'. Please rank the sections in the SI as they are chronologically referred to in the text.

Reviewer #3

(Remarks to the Author)

The authors have sufficiently addressed all my review comments.

Dr. André Dallmann
External Editor
Communications Chemistry
The Campus
4 Crinan Street
London N1 9XW
UK

John Myers
Department of Biosignals
Physikalisch-Technische Bundesanstalt
Abbestraße 2-12
10587 Berlin

31st October 2025

Dear Dr. André Dallmann,

We have revised manuscript: COMMSCHEM-25-0597 in accordance with the requests of the reviewers. We would like to thank the reviewers for their constructive comments, which we found to be helpful in the betterment of the manuscript. The changes are marked in red in the revised manuscript.

Reviewer 1

Major Concern 1.

The authors have at their disposal a NMR setup that was used to find the B_{hyp} giving the best signal. I would have expected a measurement of T_1 and T_b , using NMR for direct comparison with the new method. Why didn't they do so? I don't think time and SNR is a problem, especially for the hyperpolarized T_b . They cite reference 31, but this is not exactly in the same conditions.

Response

We would like to thank the reviewer for pointing out that by using the same setup as applied in the manuscript, it is also possible to perform NMR measurements to characterize both T_1 and T_B . We have also characterized the T_1 and T_B of $[1-^{13}\text{C}]$ pyruvate by NMR, using our setup. These data are now included in the manuscript in Section 2.3 and Figures 5b and 5c, where discussion is performed in Section 3.2 and the methodologies used, explained in Sections 4.2.1 and 4.3.2. Briefly, the T_B of the ^{13}C nucleus was found to be either 37.0 s or 26.0 s with 95% confidence intervals of [33.5 s, 41.4 s] and [21.0 s, 34.2 s], respectively depending on how the exponential fit was applied. The T_1 was found to be 43.3 s with a 95% confidence interval of [38.2 s, 50.2 s]. The T_1 results are wholly consistent with the unperturbed direct detection data presented in the manuscript, while the T_B result, depends strongly on the value attained after polarizing the sample for 180 s. This is discussed at length in Section 3.2.

Major Concern 2.

In the introduction, the authors point out the limitations of measuring T_1 and T_B of HP signals using NMR and small flip-angles. Although disentangling the effect of RF pulsing from T_1 is not as hard as the authors say (please tone down the sentence at line 144), they have a point. I am not a SQUID expert, but signal intensity is affected by the rate

at which the magnetization builds up. After all, we induce a magnetic flux. How do the authors take this phenomenon into account? In an exponential growth, the signal varies fast at the beginning and almost doesn't at the end. For a relaxation is the opposite. This can be clearly seen in figure 4, where the SNR is much higher at the beginning of the polarization/relaxation. How this can affect the accuracy of the measurement? And if this is not a problem, please explain why and within which limits (*i.e.* max detectable rate without errors).

Response

We would like to thank the reviewer for their point about line 144, regarding the difficulty of having to separate the effect of the small-angle RF pulsing from spin-lattice relaxation. It was never intended to imply that this would require anything more than application of algorithms, such as the trust region reflective algorithm, which is standard practice for many data handling applications. To this end, we have revised the relevant sentence from line 144 (now line 146) to de-emphasize how the data handling is performed:

Original:

‘... this imposes additional challenges. Extraction of T_1 from the data becomes less straightforward, requiring fitting multiple parameters, using algorithms for non-linear systems.’

Revised:

‘... this imposes the additional procedure of also having to account for the effect of the repeated radio frequency pulses.’

On the topic of the second point of discussion here, regarding the effect of the rate of change of magnetic flux on signal intensity, the reviewer in fact brings up the key reason why a SQUID was chosen to perform detection for this study. Unlike the more commonly used Faraday coils used for most NMR studies, the SQUID can be operated to output a linear response with magnetic flux, rather than with the rate of change of magnetic flux ($S \propto \Phi$, where S is the SQUID output and Φ the magnetic flux). This specifically refers to the flux-locked loop operation mode, which was applied for the results from this study.

To avoid any potential confusion, starting from line 193, this is addressed by changing:

This limitation can be circumvented through use of a highly sensitive sensor, such as a superconducting quantum interference device (SQUID) (Figure 2a) or an optically pumped magnetometer (OPM). These sensors are able to detect near-DC signals on the order of ~ 1 pT or higher (Figure 2b), which is about what would be expected from the hyperpolarization of $[1-^{13}\text{C}]$ pyruvate by SABRE-SHEATH [26].

to:

Additionally, there is also the general challenge of detecting a DC signal. The oft employed Faraday coil for NMR experiments measures the rate of change of magnetic flux, and thus cannot detect DC signals. This can be circumvented through use of alternative magnetometers, such as a superconducting quantum interference device (SQUID) (Figure 2a) or an optically pumped magnetometer (OPM). Since these sensors measure flux density rather than the rate of change of magnetic flux, they are able to detect DC signals and are capable of detecting signals on the order of ~ 1 pT or higher (Figure 2b), which is about what would be expected from the hyperpolarization of $[1-^{13}\text{C}]$ pyruvate by SABRE-SHEATH [29].

Regarding the comment about Figure 4, the features mentioned by the reviewer are not a result of the different rates of the change in flux. The greater variation observed at the end of the buildup phase versus the relaxation phase is a result of the bubbling pH_2 , which is constantly changing the sample to detector distance. This effect is alluded to in Section 3.3 of the discussion, which examines the role of gas-fluid dynamics on the sample setup.

Addressing the last part of the comment from the reviewer with regards to the max detectable rate without errors, since the signal is not dependent on the rate of change of magnetic flux, the whole exponential growth and decay are faithfully reproduced over the entire frequency range.

Minor Comments

1. line 162: please provide name and references of other HP techniques
 - **Response:** The following sentence has been added: ‘These include methods based parahydrogen [14], dynamic nuclear polarization [15] and the nuclear Overhauser effect [16] among many others.’
2. line 171: does T_1 s stand for T_1 -SABRE? If so please specify
 - **Response:** It does not. To avoid confusion, T_1 s from line 171 (new line 174) has been changed to ‘spin-lattice relaxation times’ to avoid any ambiguity.
3. line 172: 100s is a typo
 - **Response:** This was not a typo, but to avoid confusion, the text has been changed from: ‘Using fields on the order of 100s nT to μT s...’ to: ‘Using fields on the order of hundreds of nT to some μT ...’ in line 176.
4. line 176: remove human studies from a sentence where SABRE appears. SABRE, so far, has not been demonstrated for humans. It could be misleading for people not from the field
 - **Response:** A fair point. The sentence from this line has been changed from: ‘Additionally, Iali et al. were also able to demonstrate the hyperpolarization of ^{13}C pyruvate, a metabolite key to dDNP studies on human metabolism and cancer, using SABRE-SHEATH, making it also a molecule of interest for unperturbed direct detection [21].’ to: ‘One such molecule of interest for dDNP studies on human metabolism and cancer is ^{13}C pyruvate. In 2019, Iali et al. were also able to demonstrate its hyperpolarization, using SABRE-SHEATH, making it also a molecule of interest for unperturbed direct detection [24].’
5. line 187: for better understanding, it would be appropriate to provide an estimation of B_μ in case a preclinical dose (70 mM) of pyruvate is polarized to let’s say 10%
 - **Response:** From the interpretation that the reviewer is asking for an estimation of B_μ in practice from a preclinical setting, this is not so straightforward to provide, since the magnitude of B_μ relies not only upon spin density and polarization, but also on sample geometry and positioning between the sample and detector. This is addressed fully in previously published work in Myers et al. Characterization of nuclear magnetism at ultralow and zero field using SQUIDS. *IEEE Transactions on Applied Superconductivity* 1–5 (2025). It is for this reason that in line 187 (new line 193), the words ‘typically over 10 000× smaller’ are used in the introduction, due to these dependencies. For the setup applied here, given 10% polarization of 70 mM $[1-^{13}\text{C}]$ pyruvate, we would detect a B_μ of 217 pT, which we now quote on line 257 of Section 2.2.

Reviewer 2

Major Concerns

The reviewer states:

Although the level of scientific work, with the hardware development, the repeatable data at zero field, and efforts in noise reduction are remarkable, the manuscript does not convincingly demonstrate a significant advancement. Since optimal field conditions can already be predicted through level anti-crossing simulations or empirically determined via field cycling in existing platforms, such as those developed by Michael Tyler, James Eills, and collaborators using an OPM detector, the added value of the current approach remains unclear, as similar conclusions could likely be drawn using these alternative methods.

Response

The reviewer expresses concerns about the significance of advancements demonstrated in this work. We would like to posit that for liquid NMR at ultra-low field, unperturbed detection of longitudinal magnetization has not yet been demonstrated before this work. With regards in particular to work from Tayler *et al.*, they performed their work with an OPM. Here, we show how SQUIDs can also be implemented for detecting longitudinal detection. Additionally, unlike Tayler, we show that the ultra-high broadband sensitivity of SQUID detectors makes this possible without having to apply successive π pulses. Additionally, we verify the findings of our direct detection measurements with comparison to NMR on the same setup. Lastly, the reviewer also mentions the use of simulations for predicting optimal field conditions for SABRE-SHEATH. We would like to put forward that the work from the Warren group shows that we still have gaps in our understanding of SABRE-SHEATH hyperpolarization, necessitating innovative, empirical techniques such as the one we present in our work.

Minor Concerns

1. In the abstract ‘Here, we demonstrate the unperturbed, direct detection of nuclear magnetism, using a SQUID-based detector with frequency-independent sensitivity, enabling perturbation-free measurement. To compensate for low sensitivity at ultra-low magnetic fields, [1-13C]pyruvate was hyperpolarized with the SABRE-SHEATH technique at 150 nT and 500 nT’ The abstract could more clearly explain the rationale for using ultralow field, especially for non-specialists. Rephrasing to introduce that SQUID detectors are well-suited for unperturbed detection at low fields would help clarify the motivation.
 - **Response:** We agree with the reviewer’s point that the abstract could be made more accessible to the non-expert by bringing up the rationale behind ultra-low fields and SQUID detectors, while still remaining within the strict word limit. Therefore, we have completely rewritten the abstract to in order to explain more of the rationale behind ultra-low fields while remaining under 150 words.
2. On Figure 2, figure 2 the 5° tilt angle from B0 should be explained in the legend, or at least the methods sections should be referred in the legend. The SABRE-SHEATH reactor should be clearly labeled in the figure, using a color scheme accessible to colorblind readers.
 - **Response:** We thank the review for their concerns regarding Figure 2. To address them, we have added the description, ‘The SABRE reactor and SQUID dewar are tilted by 5°, as shown in the figure to aid backflow of sample upon bubbling.’ into the figure caption. We have also circled the SABRE reactor within the figure and have now included a picture of the setup in the SI (Figure S1).

3. In Figure 4, it is unclear why the polarization does not return to the baseline or why the signal intensity varies between cycles. A brief explanation of these variations and a comment on repeatability would strengthen the interpretation.
 - **Response:** Figure 4 shows the direct detection results prior to demeaning of each complete individual buildup-relaxation cycle. Therefore, there is still some residual magnetic field drift inside of the MSR that causes each cycle to have a slightly different baseline. This effect is random and is minimized upon averaging, which leads to Figure 5a. To avoid confusion, the following line has been added to the third paragraph of Section 4.4 on line 638: ‘For a single 30 min. acquisition at $B_{\text{Hyp}} = 500$ nT, the result after baseline correction, but before demeaning of individual buildup-relaxation cycles is plotted (Figure 4).’
4. The software to treat the data and perform the fittings should be indicated in the Methods section.
 - **Response:** Done
5. A supplementary data material would be useful
 - **Response:** Per the reviewer’s suggestion, we have created an SI to go with this manuscript.
6. The manuscript should include the fitting equations used for build-up and relaxation curves and report uncertainties in the fitted parameters.
 - **Response:** We have included all fitting equations for the buildup and relaxation curves in the SI (Section S3). However, for the direct detection data, we have omitted the uncertainty of the fit in the main manuscript, since it is smaller than what we report T_B and T_1 to be. For the NMR data requested by Reviewer 1, measuring both T_B and T_1 , we have included the 95% confidence interval for the fitting parameters.
7. It would have been nice to have a picture of the setup
 - **Response:** We understand the reviewer’s point here on the utility of visualising the setup from the schematic with a picture. Therefore, we have now included a picture of the setup in the SI (Figure S1).
8. The methodology for polarization quantification should be clearly described, as in the ArXiv preprint.
 - **Response:** We agree with the reviewer’s point that since the polarization quantification methodology applied was non-standard to this field, it is sensible to include more detail on how it was performed. To this end, we have included the following sentence in Section 4.3.1: ‘Final sample polarization was calculated by discretizing the sample volume of the SABRE reactor into voxels and then using the principle of reciprocity to determine the total magnetic flux seen by the setup. The full procedure is as described in Ref [29].’ Since this technique has already been fully explained in a prior publication, we leave the reader with a reference to it.
9. Line 104: ‘which is the most common state probed’ can be removed
 - **Response:** We thank the reviewer for their suggestion here. However, the intent of the relevant phrase from line 104 is to direct the reader to the fact that not only systems at thermal equilibrium can be probed with NMR. This phrase is then built upon by the paragraph that starts at line 160.
10. Line 106-108: please define Boltzmann distribution with the constant k_B and T

- **Response:** This is now addressed in lines 106-108.
11. Line 121: please avoid two consecutive parenthesis for smoother reading.
 - **Response:** This has been addressed on line 123 by changing: ‘... (along the xy -plane)($\pi/2$ pulse)...’ to: ‘... (a $\pi/2$ pulse along the xy -plane)...’.
 12. Line 127: please use ‘time’ instead of ‘constant’
 - **Response:** Done
 13. Line 172 and 201: please write in letter ‘hundreds’ instead of ‘100s’
 - **Response:** Please see the response to Minor Concern 3 from Reviewer 1 for line 172. For line 201 (new line 211), the phrase: ‘...relaxation of $[1-^{13}\text{C}]$ pyruvate at 100s nT fields in real-time...’ has been changed to: ‘...relaxation of $[1-^{13}\text{C}]$ pyruvate at fields on the order of hundreds of nT in real-time...’.

Reviewer 3

Minor Concerns

1. Between figure 4 and 5 the authors normalize repeated T1 decay measurements to the maximum polarization achieved during the SABER polarization. However, T1 decay seems to converge to different values for different repetitions as seen in figure 4. Was there some additional data processing done to normalize the lower end of the T1 decay between figures 4 and 5?
 - **Response:** Between figures 4 and 5, each complete buildup-relaxation cycle was individually demeaned before the cycles were averaged together. This is described in Section 4.3.3 from line 618. Please also see the response to reviewer 2, minor concern 3.
2. The exponential fit does not qualitatively appear to perform well on the 2nd 500 nT polarization S2. The fit underestimates the value initially and overestimates it towards the end. Can the authors comment on why this fit appears to have performed worse compared to the first run?
 - **Response:** The samples, S1 and S2 were not only prepared independently of one another, but the experiments were even also carried out in different SABRE reactors. All of these factors added to increased sample variability. Specifically, S2 had a lower signal than S1 that in addition to a slightly higher vibrational noise that upon normalization for Figure 5a, increased the appearance of this effect. Due to the lower SNR, this makes the exponential model for S2 appear significantly worse than it does for S1. This is evidenced by the model appearing better for S2 at a B_{Hyp} of 150 nT, where the vibrational noise is lower. This is now addressed in Section 3.3 in the discussion, where the dependency of vibrational noise on field is shown in Figure S3.
3. The authors need to comment on how the increase in B presumably from geometric sample changes related to gas bubbling were managed during exponential fitting. Where those values not included in the fits, was the initial value for the fit greater than 1?
 - **Response:** The values prior to the minimum of the average buildup and the values prior to the maximum value of the average relaxation were not included in the exponential fit. To make this clear, the following sentence has been added to line 618 of Section 4.3.3: ‘Any lingering effects of magnetic field drift were removed by individually demeaning each complete buildup-relaxation cycle prior to averaging. T_1 and T_B were calculated by fitting first order exponential functions to the averaged buildup and relaxation data sets, removing

the first 5 s from the fitting to account for time to allow the system to settle after starting or ceasing flow of pH_2 '

We would like to thank the reviewers again for their constructive comments and hope are changes meet their satisfaction.

Yours sincerely,
John Myers

Dr. Huijuan Guo
Senior Editor
Communications Chemistry
Heidelberger Platz 3
14197 Berlin
Germany

John Myers
Department of Biosignals
Physikalisch-Technische Bundesanstalt
Abbestraße 2-12
10587 Berlin

27th November 2025

Dear Dr. Huijuan Guo,

We have revised manuscript: COMMSCHEM-25-0597 in accordance with the requests of the reviewers and the editorial team. We would like to thank the reviewers and editors for their constructive comments, which we found to be helpful in the betterment of the manuscript. The changes are marked in red in the revised manuscript.

Reviewer 2

Minor Comments

1. Line 37 : limits \Rightarrow limit
 - **Response:** Fixed
2. Line 195 : oft or often ?
 - **Response:** ‘oft’ here is correct. To remove ambiguity a hyphen has been added between ‘oft’ and ‘employed’ to emphasize its usage better.
3. Lines 338, 354, 499, 595: please refer to the dedicated section or figure in the SI. Example for line 499, please refer to Figure or Section S1 in the SI instead of ‘see SI for a picture’. Please rank the sections in the SI as they are chronologically referred to in the text.
 - **Response:** Done

Additional Changes

1. One line 209, the phrase ‘to the best of our knowledge’ was added after ‘for the first time’ to soften the statement.
2. A data availability statement was added after the methods section (Section 5).
3. Titles were given to all figures.

We would like to thank the reviewers and editors again for their constructive comments and hope our changes meet their satisfaction.

Yours sincerely,
John Myers